# Generating Long Videos of Dynamic Scenes

**Tim Brooks**
NVIDIA, UC Berkeley

**Janne Hellsten**
NVIDIA

**Miika Aittala**
NVIDIA

**Ting-Chun Wang**
NVIDIA

**Timo Aila**
NVIDIA

**Jaakko Lehtinen**
NVIDIA, Aalto University

**Ming-Yu Liu**
NVIDIA

**Alexei A. Efros**
UC Berkeley

**Tero Karras**
NVIDIA

## Abstract

We present a video generation model that accurately reproduces object motion, changes in camera viewpoint, and new content that arises over time. Existing video generation methods often fail to produce new content as a function of time while maintaining consistencies expected in real environments, such as plausible dynamics and object persistence. A common failure case is for content to never change due to over-reliance on inductive biases to provide temporal consistency, such as a single latent code that dictates content for the entire video. On the other extreme, without long-term consistency, generated videos may morph unrealistically between different scenes. To address these limitations, we prioritize the time axis by redesigning the temporal latent representation and learning long-term consistency from data by training on longer videos. We leverage a two-phase training strategy, where we separately train using longer videos at a low resolution and shorter videos at a high resolution. To evaluate the capabilities of our model, we introduce two new benchmark datasets with explicit focus on long-term temporal dynamics.

## 1   Introduction

Videos are data that change over time, with complex patterns of camera viewpoint, motion, deformation and occlusion. In certain respects, videos are unbounded — they may last arbitrarily long and there is no limit to the amount of new content that may become visible over time. Yet videos that depict the real world must also remain consistent with physical laws that dictate which changes over time are feasible. For example, the camera may only move through 3D space along a smooth path, objects cannot morph between each other, and time cannot go backward. Generating long videos thus requires the ability to produce endless new content while maintaining appropriate consistencies.

In this work, we focus on generating long videos with rich dynamics and new content that arises over time. While existing video generation models can produce "infinite" videos, the type and amount of change along the time axis is highly limited. For example, a synthesized infinite video of a person talking will only include small motions of the mouth and head. Moreover, common video generation datasets often contain short clips with little new content over time, which may inadvertently bias the design choices toward training on short segments or pairs of frames, forcing content in videos to stay fixed, or using architectures with small temporal receptive fields.

We make the time axis a first-class citizen for video generation. To this end, we introduce two new datasets that contain motion, changing camera viewpoints, and entrances/exits of objects and scenery over time. We learn long-term consistencies by training on long videos and design a temporal latent representation that enables modeling complex temporal changes. Figure 1 illustrates the rich motion and scenery changes that our model is capable of generating. See our webpage[1] for video results, code, data and pretrained models.

---

[1] `https://www.timothybrooks.com/tech/long-videos`

36th Conference on Neural Information Processing Systems (NeurIPS 2022).

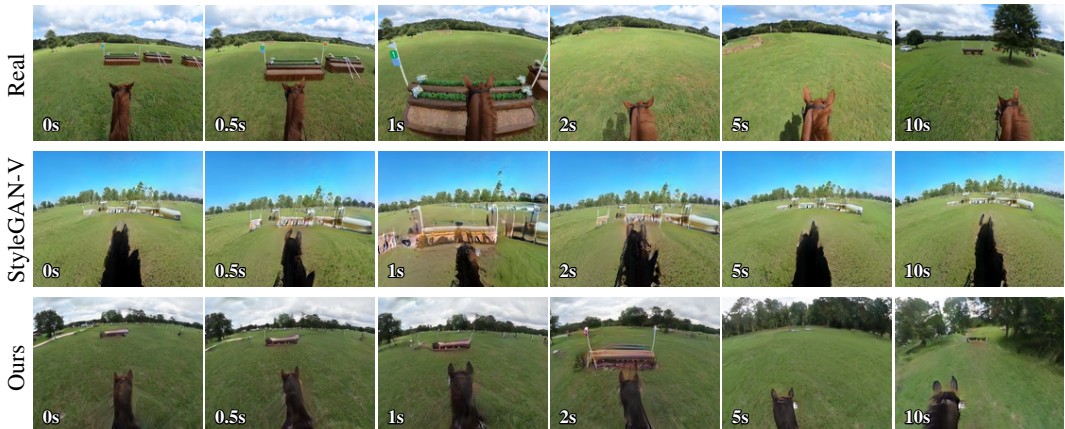

Figure 1: We aim to generate videos that accurately portray motion, changing camera viewpoint, and new content that arises over time. **Top:** Our horseback riding dataset exhibits these types of changes as the horse moves forward in the environment. **Middle:** StyleGAN-V, a state-of-the-art video generation baseline, is incapable of generating new content over time; the horse fails to move forward past the obstacle, the scene does not change, and the video morphs back and forth within a short window of motion. **Bottom:** Our novel video generation model prioritizes the time axis and generates realistic motion and scenery changes over long durations. The same videos can be viewed on the supplemental webpage.

Our main contribution is a hierarchical generator architecture that employs a vast temporal receptive field and a novel temporal embedding. We employ a multi-resolution strategy, where we first generate videos at low resolution and then refine them using a separate super-resolution network. Naively training on long videos at high spatial resolution is prohibitively expensive, but we find that the main aspects of a video persist at a low spatial resolution. This observation allows us to train with long videos at low resolution and short videos at high resolution, enabling us to prioritize the time axis and ensure that long-term changes are accurately portrayed. The low-resolution and super-resolution networks are trained independently with an RGB bottleneck in between. This modular design allows iterating on each network independently and leveraging the same super-resolution network for different low-resolution network ablations.

We compare our results to several recent video generative models and demonstrate state-of-the-art performance in producing long videos with realistic motion and changes in content. Code, new datasets, and pre-trained models on these datasets will be made available.

## 2   Prior work

Video generation is a challenging problem with a long history. The classic early works, Video Textures [50] and Dynamic Textures [10], model videos as textures by analogy with image textures. That is, they explicitly assume the content to be stationary over time, e.g., fire burning, smoke rising, foliage falling, pendulum swinging, etc., and use non-parametric [50] or parametric [10] approaches to model that stationary distribution. Although subsequent video synthesis works have dropped the "texture" moniker, much of the limitations remain similar — short training videos and models which produce little or no new objects entering the frame during the video. Below we summarize some of the more recent efforts on video generation.

**Unconditional video generation.**   Many video generation works are based on GANs [14], including early models that output fixed-length videos [1, 47, 60] and approaches that use recurrent networks to produce a sequence of latent codes used to generate frames [9, 12, 55, 56]. MoCoGAN [56] explicitly disentangles "motion" from "content" and keeps the latter fixed over the entire generated video. StyleGAN-V [52] is a recent state-of-the-art model we use as a primary baseline. Similar to MoCoGAN, StyleGAN-V employs a global latent code that controls content of an entire video. MoCoGAN-HD [55], which we also compare with, and StyleVideoGAN [12] attempt to generate videos by navigating the latent space of a pretrained StyleGAN2 model [29], but struggle to produce

realistic motion. Unlike previous StyleGAN-based [28] video models, we prioritize the time axis in our generator through a new temporal latent representation, temporal upsampling, and spatiotemporal modulated convolutions. We also compare with DIGAN [66] that employs an implicit representation to generate the video pixel by pixel.

Transformers are another class of models used for video generation [13, 42, 61, 65]. We compare with TATS [13] that generates long unconditional videos with transformers, improving upon VideoGPT [65]. Both TATS and VideoGPT employ a GPT-like autoregressive transformer [4] that represents videos as sequences of tokens. However, the resulting videos tend to accumulate error over time and often diverge or change too rapidly. The models are also expensive to train and deploy due to their autoregressive nature over time and space. In concurrent work, promising results in generating diverse videos have also been demonstrated using diffusion-based models [20].

**Conditional video prediction.** A separate line of research focuses on predicting future video frames conditioned on one or more real video frames [3, 23, 34, 36, 39, 41] or past frames accompanied by an action label [6, 15, 30, 31]. Some video prediction methods focus specifically on generating infinite scenery by conditioning on camera trajectory [37, 44] and/or explicitly predicting depth [2, 37] to then simulate a virtual camera flying through a 3D scene. Our goal, on the other hand, is to support camera movement as well as moving objects by having the scene structure emerge implicitly.

**Multi-resolution training.** Training at multiple scales is a common strategy for image generation models [7, 25, 43, 46, 58]. Transformer-based video generators also employ a related two-phase setup [65, 13]. Saito *et al.* [48] subsample frames at higher resolutions in their video generator architecture to improve efficiency. A similar idea is also used in SlowFast [11] networks where different network pathways are used for high and low frame rate video streams. Acharya *et al.* [1] propose a multi-scale GAN for video generation that increases both spatial resolution and sequence length during training to produce a fixed-length video. In contrast, our multi-resolution approach is designed to enable generating arbitrarily long videos with rich long-term dynamics by leveraging training of long sequences at low resolution.

## 3 Our method

Modeling the long-term temporal behavior observed in real videos presents us with two main challenges. First, we must use long enough sequences during training to capture the relevant effects; using, e.g., pairs of consecutive frames fails to provide meaningful training signal for effects that occur over several seconds. Second, we must ensure that the networks themselves are capable of operating over long time scales; if, e.g., the receptive field of the generator spans only 8 adjacent frames, any two frames taken more than 8 frames apart will necessarily be uncorrelated with each other.

Figure 2a shows the overall design of our generator. We seed the generation process with a variable-length stream of temporal noise, consisting of 8 scalar components per frame drawn from i.i.d. Gaussian distribution. The temporal noise is first processed by a *low-resolution generator* to obtain a sequence of RGB frames at $64^2$ resolution that are then refined by a separate *super-resolution network* to produce the final frames at $256^2$ resolution.[2] The role of the low-resolution generator is to model major aspects of the motion and scene composition, which necessitates strong expressive power and a large receptive field over time, whereas the super-resolution network is responsible for the more fine-grained task of hallucinating the remaining details.

Our two-stage design provides maximum flexibility in terms of generating long videos. Specifically, the low-resolution generator is designed to be fully convolutional over time, so the duration and time offset of the generated video can be controlled by shifting and reshaping the temporal noise, respectively. The super-resolution network, on the other hand, operates on a frame-by-frame basis. It receives a short sequence of 9 consecutive low-resolution frames and outputs a single high-resolution frame; each output frame is processed independently using a sliding window. The combination of fully-convolutional and per-frame processing enables us to generate arbitrary frames in arbitrary order, which is highly desirable for, e.g., interactive editing and real-time playback.

---

[2]We handle datasets with non-square aspect ratio by shrinking all intermediate data accordingly. With $256 \times 144$ target resolution, for example, the low-resolution frames will have $64 \times 36$ resolution.

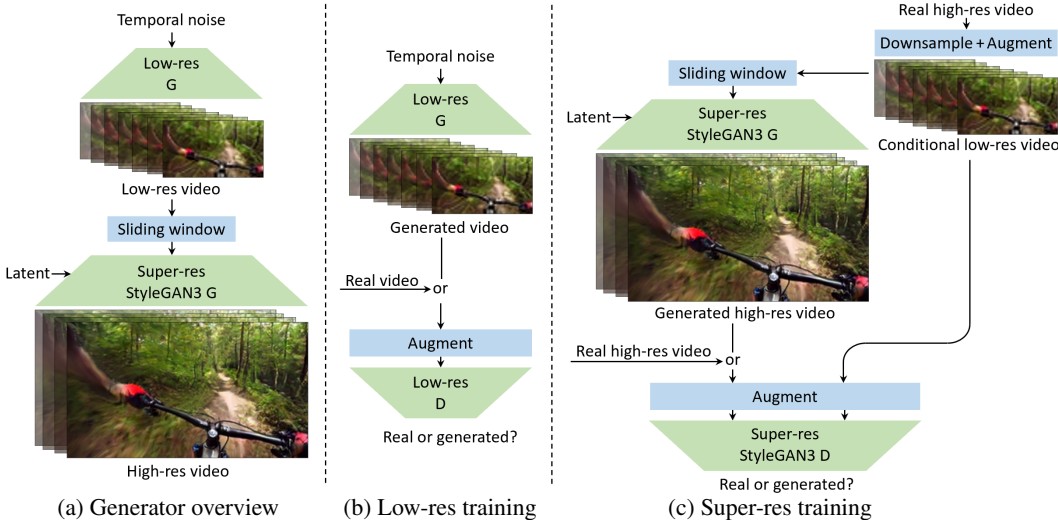

| (a) Generator overview | (b) Low-res training | (c) Super-res training |

Figure 2: Overview of our method. **(a)** To achieve long temporal receptive field and high spatial resolution, we split our generator into two components: a low-resolution generator, responsible for modeling major aspects of the motion and scene composition, and a super-resolution network, responsible for hallucinating fine details. **(b)** The low-resolution generator (Section 3.1) employs a wide temporal receptive field and is trained with sequences of 128 frames at $64^2$ resolution. **(c)** The super-resolution network (Section 3.2) is conditioned on short sequences of low-resolution frames and trained to produce their plausible counterparts at $256^2$ resolution.

The low-resolution and super-resolution networks are modular with an RGB bottleneck in between. This greatly simplifies experimentation, since the networks are trained independently and can be used in different combinations during inference. We will first describe the training and architecture of the low-resolution generator in Section 3.1 and then discuss the super-resolution network in Section 3.2.

## 3.1 Low-resolution generator

Figure 2b shows our training setup for the low-resolution generator. In each iteration, we provide the generator with a fresh set of temporal noise to produce sequences of 128 frames (4.3 seconds at 30 fps). To train the discriminator, we sample corresponding sequences from the training data by choosing a random video and a random interval of 128 frames within that video.

We have observed that training with long sequences tends to exacerbate the issue of overfitting [26]. As the sequence length increases, we suspect that it becomes harder for the generator to simultaneously model temporal dynamics at multiple time scales, but at the same time, easier for the discriminator to spot any mistakes. In practice, we have found strong discriminator augmentation [26, 69] to be necessary in order to stabilize the training. We employ DiffAug [69] using the same transformation for each frame in a sequence, as well as fractional time stretching between $\frac{1}{2}\times$ and $2\times$; see Appendix C.1 for details.

**Architecture.** Figure 3 illustrates the architecture of our low-resolution generator. Our main goal is to make the time axis a first-class citizen, including careful design of a temporal latent representation, temporal style modulation, spatiotemporal convolutions, and temporal upsamples. Through these mechanisms, our generator spans a vast temporal receptive field (5k frames), allowing it to represent temporal correlations at multiple time scales.

We employ a style-based design, similar to Karras *et al.* [29, 27], that maps the input temporal noise into a sequence of *intermediate latents* $\{w_t\}$ used to modulate the behavior of each layer in the main synthesis path. Each intermediate latent is associated with a specific frame, but it can significantly influence the scene composition and temporal behavior of several frames through hierarchical 3D convolutions that appear in the main path.

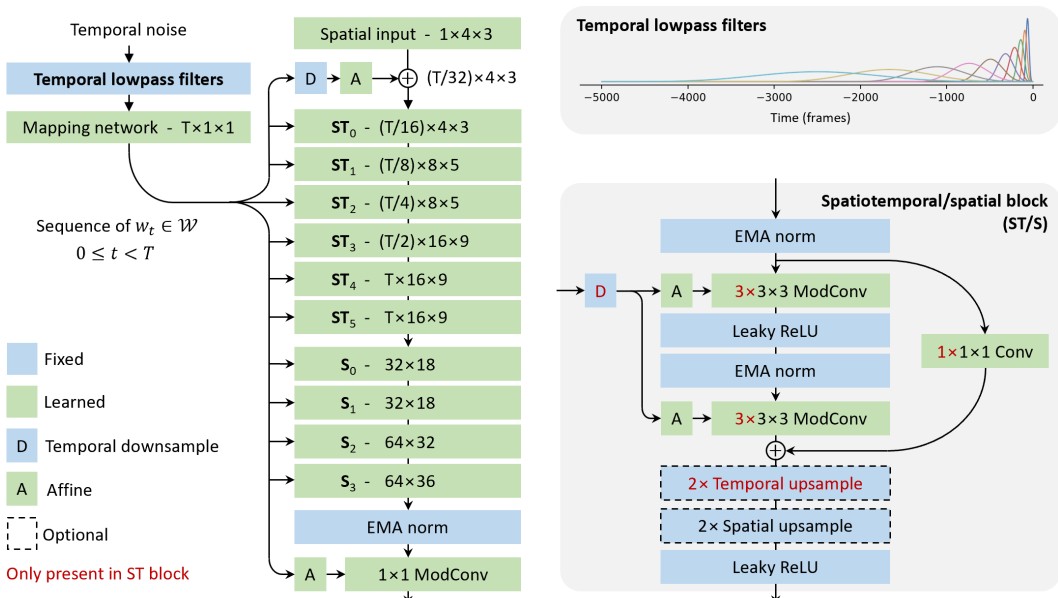

Figure 3: Low-resolution generator architecture, illustrated for 64×36 output. **Left:** The input temporal noise is mapped to a sequence of *intermediate latents* $\{w_t\}$ that modulate the intermediate activations of the main synthesis path. **Top right:** To facilitate the modeling of long-term dependencies, we enrich the temporal noise by passing it through a series of lowpass filters whose temporal footprints range all the way from 100 to 5000 frames. **Bottom right:** The main synthesis path consists of *spatiotemporal* (ST) and *spatial* (S) blocks that gradually increase the resolution over time and space.

In order to reap the full benefits of the style-based design, it is crucial for the intermediate latents to capture long-term temporal correlations, such as weather changes or persistent objects. To this end, we adopt a scheme where we first enrich the input temporal noise using a series of temporal lowpass filters and then pass it through a fully-connected *mapping network* on a frame-by-frame basis. The goal of the lowpass filtering is to provide the mapping network with sufficient long-term context across a wide range of different time scales. Specifically, given a stream of temporal noise $z(t) \in \mathbb{R}^8$, we compute the corresponding enriched representation $z'(t) \in \mathbb{R}^{128 \times 8}$ as $z'_{i,j} = f_i * z_j$, where $\{f_i\}$ is a set of 128 lowpass filters whose temporal footprint ranges from 100 to 5000 frames, and $*$ denotes convolution over time; see Appendix C.2 for details.

The main synthesis path starts by downsampling the temporal resolution of $\{w_t\}$ by 32× and concatenating it with a learned constant at $4^2$ resolution. It then gradually increases the temporal and spatial resolutions through a series of processing blocks, illustrated in Figure 3 (bottom right), focusing first on the time dimension (ST) and then the spatial dimensions (S). The first four blocks have 512 channels, followed by two blocks with 256, two with 128 and two with 64 channels. The processing blocks consist of the same basic building blocks as StyleGAN2 [29] and StyleGAN3 [27] with the addition of a skip connection; the intermediate activations are normalized before each convolution [27] and modulated [29] according to an appropriately downsampled copy of $\{w_t\}$. In practice, we employ bilinear upsampling [28] and use padding [27] for the time axis to eliminate boundary effects. Through the combination of our temporal latent representation and spatiotemporal processing blocks, our architecture is able to model complex and long-term patterns across time.

For the discriminator, we employ an architecture that prioritizes the time axis via wide temporal receptive field, 3D spatiotemporal and 1D temporal convolutions, and spatial and temporal downsamples; see Appendix C.3 for details.

### 3.2 Super-resolution network

Figure 2c shows our training setup for the super-resolution network. Our video super-resolution network is a straightforward extension of StyleGAN3 [27] for conditional frame generation. Unlike

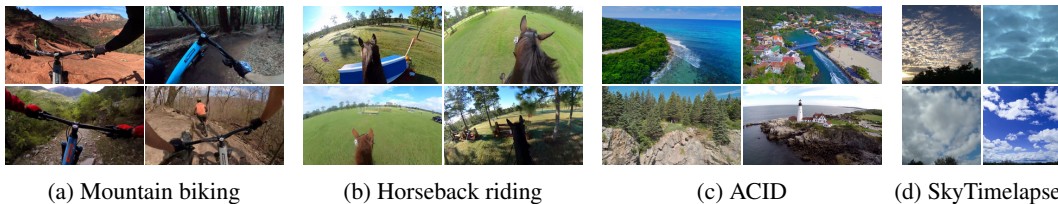

|  |  |  |  |
|---|---|---|---|
| (a) Mountain biking | (b) Horseback riding | (c) ACID | (d) SkyTimelapse |

Figure 4: Example real frames from training datasets. We introduce first-person datasets of **(a)** mountain biking and **(b)** horseback riding videos that contain complex motion and new content over time. We also evaluate on existing datasets of **(c)** nature drone footage and **(d)** sky timelapse videos.

the low-resolution network that outputs a sequence of frames and includes explicit temporal operations, the super-resolution generator outputs a single frame and only utilizes temporal information at the input, where the real low-resolution frame and 4 neighboring real low-resolution frames before and after in time are concatenated along the channel dimension to provide context. We remove the spatial Fourier feature inputs and resize and concatenate the stack of low-resolution frames to each layer throughout the generator. The generator architecture is otherwise unchanged from StyleGAN3, including the use of an intermediate latent code that is sampled per video. Low-resolution frames undergo augmentation prior to conditioning as part of the data pipeline, which helps ensure generalization to *generated* low-resolution images.

The super-res discriminator is a similar straightforward extension of the StyleGAN discriminator, with 4 low and high-resolution frames concatenated at the input. The only other change is the removal of the minibatch standard deviation layer that we found unnecessary in practice. Both low- and high-resolution segments of 4 frames undergo adaptive augmentation [26] where the same augmentation is applied to all frames at both resolutions. Low-resolution segments also undergo aggressive dropout ($p = 0.9$ probability of zeroing out the entire segment), which prevents the discriminator from relying too heavily on the conditioning signal; see Appendix D.1 for details.

We find it remarkable that such a simple video super-resolution model appears sufficient for producing reasonably good high-resolution videos. We focus primarily on the low-resolution generator in our experiments, utilizing a single super-resolution network trained per dataset. We feel that replacing this simple network with a more advanced model from the video super-resolution literature [16, 24, 49, 54] is a promising avenue for future work.

## 4 Datasets

Most of the existing video datasets introduce little or no new content over time. For example, talking head datasets [8, 45, 62, 63] show the same person for the duration of each video. UCF101 [53] portrays diverse human actions, but the videos are short and contain limited camera motion and little or no new objects that enter the videos over time.

To best evaluate our model, we introduce two new video datasets of first-person mountain biking and horseback riding (Figure 4a,b) that exhibit complex changes over time. Our new datasets include subject motion of the horse or biker, a first-person camera viewpoint that moves through space, and new scenery and objects over time. The videos are available in high definition and were manually trimmed to remove problematic segments, scene cuts, text overlays, obstructed views, etc. The mountain biking dataset has 1202 videos with a median duration of 330 frames at 30 fps, and the horseback dataset has 66 videos with a median duration of 6504 frames also at 30fps. We have permission from the content owners to publicly release the datasets for research purposes. We believe our new datasets will serve as important benchmarks for future work.

We also evaluate our model on the ACID dataset [38] (Figure 4c) that contains significant camera motion but lacks other types of motion, as well as the commonly used SkyTimelapse dataset [67] (Figure 4d) that exhibits new content over time as the clouds pass by, but the videos are relatively homogeneous and the camera remains fixed.

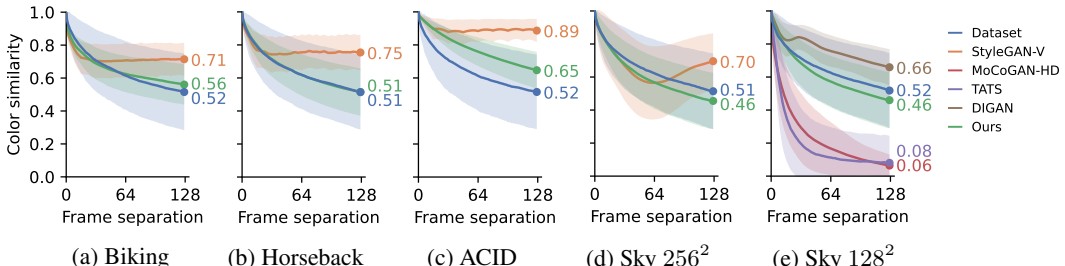

Figure 5: Color similarity (Eq. 1) of real and generated videos as a function of frame separation, reported as the mean (solid lines) and standard deviation (shaded regions) over 1000 random clips.

## 5 Results

We evaluate our model through qualitative examination of the generated videos (Section 5.1), analyzing color change over time (Section 5.2), computing the FVD metric (Section 5.3), and ablating the key design choices (Section 5.4). We compare with StyleGAN-V [52] on all datasets. Mountain biking, horseback riding and ACID [37] datasets contain videos with a 16×9 widescreen aspect ratio. We train at 256×144 resolution on these datasets to preserve the aspect ratio. Since StyleGAN-V is based on StyleGAN2 [29], we can easily extend it to support non-square aspect ratios by masking real and generated frames during training. We found it necessary to increase the R1 $\gamma$ hyperparameter by 10× to produce good results with StyleGAN-V on our new datasets that exhibit complex changes over time. We compare with MoCoGAN-HD [56], TATS [13] and DIGAN [66] using pre-trained models for the SkyTimelapse dataset at $128^2$ resolution. For these comparisons, we train a separate super-resolution network to output the frames at $128^2$ resolution, but use the same low-resolution generator as in the $256^2$ comparison.

### 5.1 Qualitative results

The major qualitative difference in results is that our model generates realistic new content over time, whereas StyleGAN-V continually repeats the same content. The effect is best observed by watching videos on the supplemental webpage and is additionally illustrated in Figure 1. Scenery changes over time in real videos and our results as the horse moves forward through space. However, the videos generated by StyleGAN-V tend to morph back to the same scene at regular intervals. Similar repeated content from StyleGAN-V is apparent on all datasets. For example, results on the webpage for the SkyTimelapse dataset show that clouds generated by StyleGAN-V repeatedly move back and forth. MoCoGAN-HD and TATS suffer from unrealistic rapid changes over time that diverge, and DIGAN results contain periodic patterns visible in both space and time. Our model is capable of generating a constant stream of new clouds.

As a further validation of our observations, we conducted a preliminary user study on Amazon Mechanical Turk. We created 50 pairs of videos for each of the 4 datasets. Each pair contained a random video generated by StyleGAN-V and one generated by our method, and we asked the participants which of them exhibited more realistic motion in a forced-choice response. Each pair was shown to 10 participants, resulting in a total of 50×4×10 responses. Our method was preferred over 80% of the time for every dataset. Please see Appendix A.1 for details.

### 5.2 Analyzing color change over time

To gain insight into how well different methods produce new content at appropriate rates, we analyze how the overall color scheme changes as a function of time. We measure color similarity as the intersection between RGB color histograms; this serves as a simple proxy for actual content changes and helps reveal the biases of different models. Let $H(x, i)$ denote a 3D color histogram function that computes the value of histogram bin $i \in [1, \dots, N^3]$ for the given image $x$, normalized so that $\sum_i H(x, i) = 1$. Given video clip $\boldsymbol{x} = \{x_t\}$ and frame separation $t$, we define the color similarity as

$$S(\boldsymbol{x}, t) = \sum_i \min \big( H(x_0, i), \ H(x_t, i) \big), \tag{1}$$

| | Biking | | Horseback | | ACID | | Sky $256^2$ | |
|---|---|---|---|---|---|---|---|---|
| | $\text{FVD}_{128}$ | $\text{FVD}_{16}$ | $\text{FVD}_{128}$ | $\text{FVD}_{16}$ | $\text{FVD}_{128}$ | $\text{FVD}_{16}$ | $\text{FVD}_{128}$ | $\text{FVD}_{16}$ |
| StyleGAN-V | 533.3 | 353.7 | 427.0 | 319.2 | 112.4 | 91.5 | 151.2 | 48.4 |
| with $10\times$ R1 $\gamma$ | 224.6 | 99.2 | 196.2 | 159.0 | – | – | – | – |
| Ours | 113.7 | 83.8 | 95.9 | 113.5 | 166.6 | 127.3 | 152.7 | 116.5 |

| | Sky $128^2$ | |
|---|---|---|
| | $\text{FVD}_{128}$ | $\text{FVD}_{16}$ |
| MoCoGAN-HD | 635.6 | 224.9 |
| TATS | 435.0 | 97.0 |
| DIGAN | 228.6 | 153.4 |
| Ours | 142.6 | 107.5 |

Table 1: We compute FVD on segments of 128 and 16 frames ($\text{FVD}_{128}$ and $\text{FVD}_{16}$ respectively), where lower is better. **Left:** Our model outperforms StyleGAN-V on horseback riding and mountain biking datasets – both of which contain complex motion and new content over time. Our model underperforms StyleGAN-V on ACID and SkyTimelapse despite qualitative improvements and favorable user study ratings in Section 5.1. **Right:** Our model outperforms MoCoGAN-HD, TATS and DIGAN baselines on SkyTimelapse at $128^2$ resolution on $\text{FVD}_{128}$.

where $S(\boldsymbol{x}, t) = 1$ indicates that the color histograms are identical between $x_0$ and $x_t$. In practice, we set $N = 20$ and report the mean and standard deviation of $S(\cdot, t)$, measured on 1000 random video clips containing 128 frames each.

Figure 5 shows $S(\cdot, t)$ as a function of $t$ for real and generated videos on each dataset. The curves trend downward over time for real videos as content and scenery gradually change. StyleGAN-V and DIGAN are biased toward colors changing too slowly — both of these models include a global latent code that is fixed over the entire video. On the other extreme, MoCoGAN-HD and TATS are biased toward colors changing too quickly. These models use recurrent and autoregressive networks, respectively, both of which suffer from accumulating errors. Our model closely matches the shape of the target curve, indicating that colors in our generated videos change at appropriate rates.

Color change is a crude approximation of the complex changes over time in videos. In Appendix A.3 we also consider LPIPS [68] perceptual distance instead of color similarly and observe the same trends in most cases.

### 5.3  Fréchet video distance (FVD)

The commonly used Fréchet video distance (FVD) [57] attempts to measure similarity between real and generated video distributions. We find that FVD is sensitive to the realism of individual frames and motion over short segments, but that it does not capture long-term realism. For example, FVD is essentially blind to unrealistic repetition of content over time, which is prominent in StyleGAN-V videos on all of our datasets. We found FVD to be most useful in ablations, i.e., when comparing slightly different variants of the same architecture.

FVD [57] computes the Wasserstein-2 distance [59] between sets of real and generated features extracted from a pre-trained I3D action classification model [5]. Skorokhodov *et al.* [52] note that FVD is highly sensitive to small implementation differences, down to the level of image compression settings, and that the reported results are not necessarily comparable between papers (Appendix C in [52]). We report all FVD results using consistent evaluation protocol, ensuring apples-to-apples comparison. We separately measure FVD using 128- and 16-frame segments, denoted by $\text{FVD}_{128}$ and $\text{FVD}_{16}$, and sample 2048 random segments from both the dataset and generator in each case.

Table 1 (left) reports FVD on all datasets for StyleGAN-V and our model. We outperform StyleGAN-V on horseback riding and mountain biking datasets that contain more complex changes over time, but underperform on ACID and slightly underperform on SkyTimelapse in terms of $\text{FVD}_{128}$. However, this underperformance strongly disagrees with the conclusions from the qualitative user study in Section 5.1. We believe this discrepancy comes from StyleGAN-V producing better individual frames, and possibly better small-scale motion, but falling seriously short in recreating believable long-term realism – and the FVD being sensitive primarily to the former aspects. Table 1 (right) reports FVD metrics on MoCoGAN-HD, TATS, DIGAN and our model for SkyTimelapse at $128^2$; we outperform all baselines in terms of $\text{FVD}_{128}$ on this comparison.

| | $FVD_{128}$ | $FVD_{16}$ |
|---|---|---|
| Ours (128 frames) | 113.7 | 83.8 |
| 16 frames | 163.6 | 108.5 |
| 2 frames | 396.8 | 169.4 |

(a) Ablation of training sequence length

| | $FVD_{128}$ | $FVD_{16}$ |
|---|---|---|
| Ours | 113.7 | 83.8 |
| 0.1× lowpass width | 153.1 | 113.2 |
| 10× lowpass width | 217.9 | 126.5 |

(b) Ablation of temporal lowpass filter footprint

Table 2: **(a)** Our model learns to generate realistic long videos by training on long videos; decreasing the sequence length used during training is consistently harmful. **(b)** The footprint of the temporal lowpass filters plays an important role in producing inputs to the low-resolution mapping network at appropriate temporal frequencies; changing the footprint by an order of magnitude hurts performance.

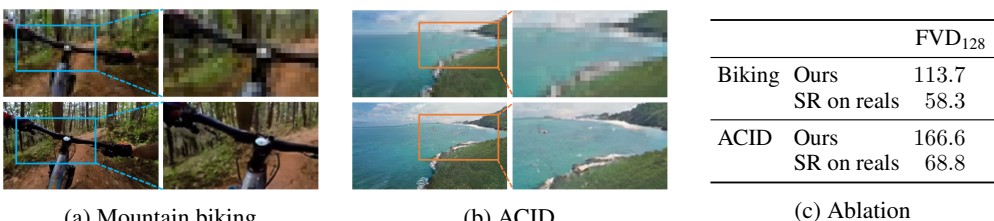

| | | $FVD_{128}$ |
|---|---|---|
| Biking | Ours | 113.7 |
| | SR on reals | 58.3 |
| ACID | Ours | 166.6 |
| | SR on reals | 68.8 |

(a) Mountain biking  (b) ACID  (c) Ablation

Figure 6: Evaluation of the super-resolution network. **(a,b)** Generated low-resolution frames and the corresponding high-resolution frames produced by the super-resolution network. **(c)** The super-resolution network yields remarkably good FVD when provided with real low-resolution videos as input; the overall quality of our results is largely dictated by the low-resolution generator.

## 5.4 Ablations

**Training on long videos improves generation of long videos.** Observing long videos during training helps our model learn long-term consistency, which is illustrated in Table 2a that ablates the sequence length used during training of the low-resolution generator. We found that the benefits of training with long videos only became evident after designing a generator architecture with appropriate temporal receptive field to utilize the rich training signal. Note that even though we ablate aspects of the low-resolution generator, we still compute FVD using the final high-resolution videos produced by the super-resolution network.

**Footprint of the temporal lowpass filters.** Our temporal latent representation serves a vital role in expanding the receptive field of our generator, modeling patterns over different time scales, and enabling the generation of new content over time. While we primarily leverage long training videos to learn long-term consistencies from data, the size of our temporal lowpass filters plays a role in encouraging the low-resolution mapping network to learn correlations at appropriate time scales. Table 2b demonstrates the negative impact of using inappropriately sized filters. We find that our model performs well with the same filter configuration for all datasets, although it is possible that the ideal settings may vary slightly between datasets.

**Effectiveness of the super-resolution network.** Figure 6a,b shows examples of low-resolution frames generated by our model along with the corresponding high-resolution frames produced by our super-resolution network; we find that the super-resolution network generally performs well. To ensure that the quality of our results is not disproportionately limited by the super-resolution network, we further measure FVD when providing the super-resolution network with *real* low-resolution videos as input in Figure 6c. Indeed, FVD greatly improves in this case, which indicates that there are still significant gains to be realized by further improving the low-resolution generator.

## 5.5 Failure cases

Separate low- and super-resolution networks makes the problem computationally feasible, but it may somewhat compromise the quality of the final high-resolution frames. We observed that "swirly" artifacts are most prominent in the super-resolution output and not in the low-resolution output. Our model also struggles with long-term consistency of small details (e.g., distant jumps in generated horseback riding videos) that begin to appear before quickly fading out. We believe these issues are due to limitations of our super-resolution network, and that improving the super-resolution network

would benefit the model in this regard. Another failure case we observed is difficulty preserving 3D consistency for scenes with very little motion, such as in the ACID dataset. In cases where there is little motion, one may consider using an explicit 3D representation.

## 6 Conclusions

Video generation has historically focused on relatively short clips with little new content over time. We consider longer videos with complex temporal changes, and uncover several open questions and video generation practices worth reassessing — the temporal latent representation and generator architecture, the training sequence length and recipes for using long videos, and the right evaluation metrics for long-term dynamics.

We have shown that representations over many time scales serve as useful building blocks for modeling complex motions and the introduction of new content over time. We feel that the form of the latent space most suitable for video remains an open, almost philosophical question, leaving a large design space to explore. For example, what is the right latent representation to model persistent objects that exit from a video and re-enter later in the video while maintaining a consistent identity?

The benefits we find from training on longer sequences open up further questions. Would video generation benefit from even longer training sequences? Currently we train using segments of adjacent frames, but it might be beneficial to use larger frame spacings to cover longer time spans.

Quantitative evaluation of the results continues to be challenging. As we observed, FVD goes only a part of the way, being essentially blind to repetitive, even very implausible results. Our tests with how the colors and LPIPS distance change as a function of time partially bridge this gap, but we feel that this area deserves a thorough, targeted investigation of its own. We hope our work encourages further research into video generation that focuses on more complex and longer-term changes over time.

**Negative societal impacts**   Our work falls within data-driven generative modeling, which, as a field, has well known potential for misuse with increasing quality improvements. The training of video generators is even more intensive computationally than training still image generators, increasing energy usage. Our project consumed 300MWh on an in-house cluster of V100 and A100 GPUs.

**Acknowledgements**   We thank William Peebles, Samuli Laine, Axel Sauer and David Luebke for helpful discussion and feedback; Ivan Skorokhodov for providing additional results and insight into the StyleGAN-V baseline; Tero Kuosmanen for maintaining compute infrastructure; Elisa Wallace Eventing (`https://www.youtube.com/c/WallaceEventing`) and Brian Kennedy (`https://www.youtube.com/c/bkxc`) for videos used to make the horseback riding and mountain biking datasets. Tim Brooks is supported by the National Science Foundation Graduate Research Fellowship under Grant No. 2020306087.

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
