# OpenReview forum: "Generating Long Videos of Dynamic Scenes"
_NeurIPS.cc/2022/Conference — NeurIPS 2022 Accept_

### Official Review · Reviewer_u8rP · 2022-07-08

**Rating:** 6
**Confidence:** 5
**Soundness:** 3 good
**Presentation:** 3 good
**Contribution:** 3 good

**Summary:**

This paper takes the initiative to improve the problem of video generation in terms of the dynamics of the generated videos. The main strategy proposed by the author is to increase the real video lengths seen by the discriminator by reducing the resolution. The reduced resolution is compensated separately with a super-resolution network.

The authors provide some analysis of the color dynamics of the real dataset and the generated videos using different methods.


**Questions:**

1 - Fig. 5 is a little hard to read due to the mix of green and blue colors. However, it seems that although the color average between the proposed method is the closest to the real data, the variance is much lower. What does that imply?

2 - I am wondering if the color similarity based on the frame separation is a good indicator/metric for better scene dynamics. Why did the authors select this metric?

3 - I found this statement confusing `time offset of the generated video can be controlled by shifting and reshaping the temporal noise, respectively.` How shifting the noise, can decide the time offset?

4 - I am wondering if a pre-trained super-resolution network (perhaps on image datasets) can perform nearly as well as what authors trained on the video examples.

5 - What about the number of generated shots? Applying a shot detector on the real vs generated data, how much the distribution of the number of shots per clip is similar per class? What about the distribution of consecutive shots similarity? Do authors think it will be a complementary indication to the color examination?


**Limitations:**

I believe the authors are honest about the negative societal impacts. I have no more points to add here.

**Strengths And Weaknesses:**

Strengths:

1- The literature has overlooked video dynamics and temporal modeling for video generation in generative models. This research has selected the correct challenge in video generation to address.

2- Great writing and supplementary materials.

3- Decoupling the temporal dynamics from frame resolution has some novelties. However, it has been remotely mentioned by previous research but has never been explicitly modeled with two networks in two different stages.

Weaknesses:

1- Although I respect the authors' braveness and choice of problem, I believe the paper conveys such a message that the main reason for low dynamics in video generation models is data. Since we are not showing long videos to the discriminator, the generated videos do not have enough dynamics or look repetitive. Although I agree with this argument, IMO, the main challenge is modeling the temporal dynamics. `We can still (wisely) select shorter videos but with enough frames change.` But we need better structures to model the temporal dynamics.

2- I appreciate the honest selection of the generated video samples in the supplementary materials. It is clear that in some scenarios, like horse riding, we see novel objects enter the scene, and the motion is not repetitive as the baselines. However, in some cases like `ACID: acid_grid.mp4, second row`,  we see a lot of distortions in the generated video. Although the method works better for high dynamics, it may not handle low dynamics.

---

> ### Author Response · Authors · 2022-08-02
> **Author response to Reviewer u8rP**
>
> Thank you for your review and insightful feedback. We are encouraged that you agree long-term dynamics is understudied and that we selected the correct challenge in video generation to address.
>
> **“Since we are not showing long videos to the discriminator, the generated videos do not have enough dynamics or look repetitive. Although I agree with this argument, IMO, the main challenge is modeling the temporal dynamics. We can still (wisely) select shorter videos but with enough frames change. But we need better structures to model the temporal dynamics.”**
>
> We agree that other approaches may be successful at modeling temporal dynamics in future work. We identified that long-term dynamics is an understudied aspect of video generation and introduced an approach using much longer videos at low resolution to improve these dynamics. As you suggest, it may be possible to instead capture long-term dynamics using fewer frames that are intelligently selected, and we will happily add discussion of this to our paper. We hope our work encourages more research on long-term dynamics and welcome other approaches.
>
> **“I appreciate the honest selection of the generated video samples in the supplementary materials. It is clear that in some scenarios, like horse riding, we see novel objects enter the scene, and the motion is not repetitive as the baselines. However, in some cases like ACID: acid_grid.mp4, second row, we see a lot of distortions in the generated video. Although the method works better for high dynamics, it may not handle low dynamics.”**
>
> Yes, there are certainly visible distortions in some results such as the video you point out, and we will mention this limitation more clearly. However, note that even for the ACID dataset with relatively low dynamics, there are unrealistic repetitions in videos generated by StyleGAN-V that our model does not suffer from. The results of our user study found that humans prefer videos generated by our model over 80% of the time on each dataset, including ACID.
>
> **“Fig. 5 is a little hard to read due to the mix of green and blue colors. However, it seems that although the color average between the proposed method is the closest to the real data, the variance is much lower. What does that imply?”**
>
> Further investigation would be needed to draw significant meaning from the standard deviation on these plots, but in general it is preferred to match the dataset. Too low of a standard deviation may indicate there is too little variation in the amount of change over time across different generated videos.
>
> **“I am wondering if the color similarity based on the frame separation is a good indicator/metric for better scene dynamics. Why did the authors select this metric?”**
>
> The color similarity plots are a simple illustration of how quickly videos change over time. They are not intended as a standalone metric of scene dynamics, but as a probe into the biases of videos generated with different models. They show when videos change far too quickly or slowly over time. We experimented with feature distances rather than color similarity in appendix A.2 and found similar results.
>
> **“I found this statement confusing 'time offset of the generated video can be controlled by shifting and reshaping the temporal noise, respectively.' How shifting the noise, can decide the time offset?”**
>
> We apologize for the confusing statement. What we meant is that because our architecture is fully-convolutional across time, the output video is translation equivariant with respect to the input temporal noise. Therefore, by translating/shifting the input temporal noise, we can translate the output video forward or backward in time. We will clarify this in the text.
>
>
> **“I am wondering if a pre-trained super-resolution network (perhaps on image datasets) can perform nearly as well as what authors trained on the video examples.”**
>
> Good question. We have not tried pretrained super-resolution networks. We did try training our super-resolution network on individual images in an early experiment, and found that it led to temporal flickering. We prioritized the low-resolution generator of long temporal dynamics, and believe replacing or improving the super-resolution module is a valuable topic for future work.
>
> **“What about the number of generated shots? Applying a shot detector on the real vs generated data, how much the distribution of the number of shots per clip is similar per class? What about the distribution of consecutive shots similarity? Do authors think it will be a complementary indication to the color examination?”**
>
> Thank you for this suggestion. We agree that comparing the distributions of detected shots per clip or similarity between shots makes sense as a potential metric. We believe determining the right evaluation for long-term video dynamics is an important topic that deserves a targeted investigation of its own.

---

### Official Review · Reviewer_3wBD · 2022-07-10

**Rating:** 7
**Confidence:** 3
**Soundness:** 3 good
**Presentation:** 4 excellent
**Contribution:** 4 excellent

**Summary:**

The paper presents a video generation model that is capable of producing new content (e.g., new object or scenery), object motion, and changes in camera viewpoint over time for longer videos than prior works. In order to achieve this, temporal modeling is emphasized in the proposed architecture, unlike existing works in which frame quality is often prioritized. As their main contribution, the authors redesign the temporal latent representation and train the carefully-designed model, which has the capability to operate over long time scales with a vast temporal receptive field, on longer videos at a low resolution and shorter videos at a high resolution. Two new benchmark datasets are introduced to best evaluate the proposed model since there are no existing datasets with long enough videos. Evaluated on 4 datasets with different characteristics, the proposed model outperforms prior methods especially qualitatively on aspects including generating plausible dynamics and object persistence, producing new content while maintaining consistencies over time.


**Questions:**

1. According to the color similarity plots, the standard deviations of Dataset are quite high, and often higher than that of models. Does it tell anything?

2. The “swirly” artifacts visible in some of the videos might be attributed to the multi-resolution strategy, and it was referred to as “RGB bottleneck” (line 305). Could you further explain this in detail?


**Limitations:**

The authors have discussed the limitations and potential negative societal impact. From the qualitative results, it seems that the model also struggles when objects in the scene interact with each other.


**Strengths And Weaknesses:**

Strength:
+ The paper has suggested several interesting insights and many useful practices for video generation. For example, a multi-resolution two-stage strategy might be a good solution for training and deployment for models to handle long videos; the low-resolution generator should be fully convolutional over time to learn long-term temporal correlations whereas the super-resolution generator can operate in a frame-by-frame basis. Videos with longer sequence length tend to exacerbate the issue of overfitting and therefore some strong augmentations might be required.

+ The paper has provided a deep investigation on the metrics of video generation. Particularly, they propose to analyze color change over time as a simple way to diagnose potential bias captured by the different models. In addition, existing commonly-used metrics such as FVD and LPIPS are also discussed, and the authors have found these metrics to agree less with the qualitative results or user study results.

+ The qualitative results are convincing. It was shown that the proposed model is capable of generating videos with rich motion and scenery changes. Existing methods are incapable of generating realistic long videos, and explanations and analysis presented in the paper are reasonable.

+ Two new datasets are proposed which might be beneficial for future researchers.



Weaknesses:
- An in-depth failure case analysis is absent, which might be interesting to have for readers.

- The proposed architecture seems to heavily depend on the correct data augmentation in use.

---

> ### Author Response · Authors · 2022-08-02
> **Author response to Reviewer 3wBD**
>
> Thank you for the thoughtful feedback. We are glad that you find our work has interesting insights and useful practices for video generation.
>
> **“An in-depth failure case analysis is absent.”**
>
> We are happy to add this in the final paper. We discuss the “swirly” artifacts below in more detail and will include that discussion. Another failure case we observed is difficulty preserving 3D consistency for scenes with very little motion, such as in the ACID dataset. In cases where there is little motion, one may consider using an explicit 3D representation to improve results in future work. A third failure case is in long-term consistency of small details (e.g., distant jumps in generated horseback riding videos) that begin to appear before quickly fading out. We believe issues with long-term small details are due to limitations of our super-resolution network, and that improving the super-resolution network in future work will address these artifacts.
>
> **“The proposed architecture seems to heavily depend on the correct data augmentation.”**
>
> Indeed, we found that using the proper strong data augmentation is especially important since we train on much longer videos that are subject to overfitting.
>
> **“According to the color similarity plots, the standard deviations of Dataset are quite high, and often higher than that of models. Does it tell anything?”**
>
> The color similarity plots illustrate bias in the rate of change over time for different models, such as if videos change far too quickly or slowly. Further investigation would be needed to draw significant meaning from the standard deviation on these plots, but in general it is preferred to match the dataset. Too low of a standard deviation may indicate there is too little variation in the amount of change over time across different generated videos.
>
> **“The “swirly” artifacts visible in some of the videos might be attributed to the multi-resolution strategy … Could you further explain this in detail?”**
>
> We observed that the “swirly” artifacts are most prominent in the super-resolution output and not in the low-resolution output. We believe the artifacts may be at least partially related to the domain gap between real and generated low-resolution videos passed as input to the super-resolution network. We mitigate this concern using low-resolution conditioning augmentation to improve generalization (see appendix D.1), but acknowledge it is still a clear area for improvement. Further investigation will be required to clarify the cause and address these artifacts in future work. We will rephrase the discussion of this limitation on line 305.

---

### Official Review · Reviewer_cBCU · 2022-07-11

**Rating:** 7
**Confidence:** 4
**Soundness:** 4 excellent
**Presentation:** 4 excellent
**Contribution:** 4 excellent

**Summary:**

Brief Summary: The paper addresses the problem of video generation with a focus on longer time horizon videos which requires consistency. To this end, the authors introduce two new benchmark datasets on horse-riding and mountain biking. The key observation is that the main components for temporal consistency are preserved at lower spatial resolution. The authors therefore propose using a hierarchical architecture to first create long low-resolution video, followed by sliding windows to create higher resolution videos at shorter time-steps.

The authors have further performed a human evaluation on mechanical turk to find their method was preferred over 80% of the time.

**Questions:**

Q1. The idea of using low and high resolution for long and small temporal segments reminds me of SlowFast networks [Ref1], where the slow path samples sparsely with more channel dimension, while fast path samples densely with less channel dimension. Obviously, it is not 1-1 correspondence, but some kind of discussion could be useful (maybe in supplementary).

[Ref1]: Feichtenhofer, Christoph, Haoqi Fan, Jitendra Malik, and Kaiming He. "Slowfast networks for video recognition." In Proceedings of the IEEE/CVF international conference on computer vision, pp. 6202-6211. 2019.

**Limitations:**

I think authors have done a good job of highlighting failure cases as well as pointing out where FVD metric is not very predictive of human eval performance.

**Strengths And Weaknesses:**

Pros:

1. The idea is simple yet elegant. Implementation-wise it is quite interesting to find that a straightforward extension of stylegan to videos where input images are simply concatenated leads to promising results.

2. New datasets contributions are always welcome. The two contributed datasets on mountain biking and horseback riding could be useful for future research especially to evaluate temporal consistency.

3. The authors have done a human eval which is very important in video generation, and found their method was preferred 80% of the time.

4. The authors also show color-change as a heuristic for temporal which clearly show the benefits of hierarchical training.

5. The visualizations provided in the supplementary are very cool!

Cons:
1. For the user study, the authors should have provided "equally same" as an option, and used sanity check that "equally same" is picked when both videos are obtained from the same model. I am not sure if this could have created any issues in the assessment (my guess is the effect could be mild, but not negligible).

2. I am wondering if a better evaluation for long-term consistency would be conditional generation, where first few and last few seconds of real video are provided. For instance, if a tree is seen at the last frame (fig-1, 10s), one could have a soft-metric that the tree be seen (albeit at lower resolution) at some intermediate frame. Could make the task on human evaluation easier as well given that they would be comparing more similar videos.

---

> ### Author Response · Authors · 2022-08-02
> **Author response to Reviewer cBCU**
>
> Thank you for your review and suggestions – we are encouraged that you find our idea elegant and the new datasets a useful contribution.
>
> **“For the user study, the authors should have provided "equally same" as an option”**
>
> For our human evaluation, we followed the forced-choice format used in established computer vision papers such as [1], and will keep your suggestion to include an "equally same" option in mind for future user studies.
>
> **“I am wondering if a better evaluation for long-term consistency would be conditional generation”**
>
> We appreciate this idea to evaluate long-term consistency using conditional generation tasks and believe determining the right evaluation for long-term video dynamics is an important topic that deserves a targeted investigation of its own.
>
>
> **“The idea of using low and high resolution for long and small temporal segments reminds me of SlowFast networks”**
>
> Thank you for this suggestion, we will happily add discussion of the related SlowFast video recognition paper.
>
>
> [1] Zhang et al., The Unreasonable Effectiveness of Deep Features as a Perceptual Metric

---

### Official Review · Reviewer_xqQ3 · 2022-07-12

**Rating:** 5
**Confidence:** 5
**Soundness:** 3 good
**Presentation:** 3 good
**Contribution:** 3 good

**Summary:**

The paper discusses a method for generating long videos in which new content is introduced as the camera moves forwards. Training on such videos especially in high resolution is prohibitively expensive. One of the possible ways to mitigate the issue is to train a two stage architecture, in which dynamics is trained on low-res, followed by super-resolving the low-res long video with the second stage. To show advantages of their approach the authors collect two new datasets, in which the camera moves forward. According to the results, the method outperformed current works on these two datasets, and finishes second on previously available datasets.

**Questions:**

Please see above.

**Limitations:**

Please see above

**Strengths And Weaknesses:**

Strengths. The paper has some:

S1. The paper shows that treating temporal noise is important. The idea of filtering it with a low-pass filter is interesting. This way only low-frequency, longer events are captured by the model.
S2. Results are quite impressive on the two new datasets. Very impressive!

There also are weaknesses, unfortunately:

W1. Subsampling in space and time has been proposed in the previous literature. It has been shown that training at high res both in space and time is prohibitively expensive. In [TGAN-V2] for example, they generate longer videos at low resolution and as they increase resolution with more generators they drop the frame rate. This way they achieved efficient training of 256^2 resolution--highest resolution of this paper--in 2018. Another similar idea was used in a discriminator of DVD-GAN [9]. They downsampled the resolution for video discriminator, and reduced frame-rate for image discriminator. [TGAN-V2] is not cited, DVD-GAN is cited but the similarities/differences are not discussed.

W2. The proposed framework is reasonable and seems to be working well on the proposed datasets. The key new interesting idea is temporal filtering, the rest of the framework from the high level is known (see W1). I acknowledge the amount of effort the authors put into making it work, of course.

W3. Other video datasets. The paper fails to report UCF, FaceForensics and others from stylegan-v & tats, arguing that these datasets contain less new content and camera movements. While it might be case, the method only reports better scores on their own proposed datasets. The problem with this is that it takes a lot of effort to tune each method to each dataset, making it not very clear if the proper and sufficient tuning for stylegan-v was made. If one compares to existing numbers on existing datasets, such as UCF, on which TATS and stylegan-v show reasonable performance, one can guarantee that the method is evaluated against the numbers in which proper time was invested. Otherwise it's always possible to select a dataset on which the method scores best.

W4. Other resolutions. Stylegan-v and mocogan-hd show training on much higher resolutions, as high as 1024. While this method focuses on the temporal part of videos, it's not clear what's the resolution upper bound. It's important to understand that even for this work, since scaling GANs is a non-trivial task both in terms of computational resources and quality.


[TGAN-V2] Saito, Masaki, et al. "Train sparsely, generate densely: Memory-efficient unsupervised training of high-resolution temporal gan." International Journal of Computer Vision 128.10 (2020): 2586-2606.

---

> ### Author Response · Authors · 2022-08-02
> **Author response to Reviewer xqQ3**
>
> We appreciate your review and feedback, and are glad you find our treatment of temporal noise interesting and the results on our new datasets impressive.
>
> **“Subsampling in space and time has been proposed in the previous literature. It has been shown that training at high res both in space and time is prohibitively expensive.”**
>
> Thank you for drawing our attention to the work of Saito et al. (TGAN-V2). We will certainly add discussion of this work and elaborate discussion of DVD-GAN. Similar to our model, these works overcame computational limits during training by using multiple resolutions and fewer frames at high resolution. TGAN-V2 and DVD-GAN used end-to-end training, and decreased framerate at higher resolutions during training in the generator and/or discriminator. Instead, we train an entirely separate video super-resolution network. This both overcame computational limits and isolated our task of generating long videos to the low resolution, enabling us to prioritize the design of a new low-resolution architecture and temporal latent space.
>
> **“The paper fails to report UCF, FaceForensics and others from stylegan-v & tats, arguing that these datasets contain less new content and camera movements. While it might be case, the method only reports better scores on their own proposed datasets.”**
>
> The focus of our work is on enabling generation of long videos (e.g., 10s) with a moving camera and new objects/scenery over time, and we believe our two new datasets were necessary to study this. We do not evaluate on the common UCF101 or FaceForensics datasets with videos that do not meet these criteria. Please note that we tuned the StyleGAN-V baseline on our datasets to be more competitive (Table 1), we evaluate on the common SkyTimelapse dataset against multiple baselines and produce competitive FVD scores, and that we outperform StyleGAN-V on all datasets including the existing SkyTimelapse and ACID datasets in terms of human preference in our user study.
>
> **“Other resolutions. Stylegan-v and mocogan-hd show training on much higher resolutions”**
>
> Indeed, we do not train at higher resolutions than 256. While a number of video generation papers have focused on high resolution output, we believe the long-term dynamics of videos with new content over time has been an understudied topic. We therefore focus primarily on the temporal aspect of video generation in this work.

---

### Meta-Review · Area_Chair_83Nx · 2022-08-24

**Recommendation:** Accept
**Confidence:** Certain

**Metareview:**

All four reviewers enjoyed this paper and were particularly impressed by the videos provided in the supplementary material. The results are very impressive indeed. The reviewers also agreed that using a multi stage approach was interesting and effective. The two new datasets were deemed useful to the generation community and the proposed metrics and human evaluations were appreciated by the reviewers. A few smaller concerns included a missing failure analysis and some clarifications questions which were addressed in the rebuttal. Given the above, I recommend acceptance.

**Award:**

No

---

### Decision · Program_Chairs · 2022-09-14

Accept